# FR-NAS: Forward-and-Reverse Graph Predictor for Efficient Neural Architecture Search

## Abstract

Neural Architecture Search (NAS) has risen to prominence as a pivotal tool for identifying optimal configurations for deep neural networks suited to particular tasks. However, the process of training and assessing numerous architectures introduces considerable computational overhead. One approach to mitigate this is through performance predictors, which offer a means to estimate an architecture's potential without exhaustive training. Given that neural architectures fundamentally resemble directed acyclic graphs (DAGs), graph neural networks (GNNs) become an apparent choice for such predictive tasks. Nevertheless, the scarcity of training data can impact the precision of GNN-based predictors. To address this, we introduce a novel GNN predictor for NAS. This predictor renders neural architectures into vector representations by combining both the conventional and inverse graph views. Additionally, we incorporate a tailored feature loss within the GNN predictor to ensure efficient utilization of both types of representations. We subsequently assess our method's efficacy through experiments on benchmark datasets including NASBench-101, NASBench-201, and the DARTS search space, with a training data range of 50 to 400 samples. The results demonstrated a notable performance improvement, achieving an enhancement of 3%-16% in terms of prediction accuracy when compared to state-of-the-art GNN predictors across the board. The source code will be made publicly available.

## 1 Introduction

Neural Architecture Search (NAS) plays a pivotal role in the automated generation of high-performing deep neural networks. Given a designated train and test dataset, NAS can be viewed as an optimization problem which aims to discover an architecture that maximizes accuracy and other performance metrics within a circumscribed search domain. Typically, NAS algorithms derive feedback from the performance of numerous probed architectures, facilitating the generation of superior architectural configurations. A number of approaches have been adopted have been judiciously deployed to effectuate NAS tasks, including Reinforcement Learning (Tan et al., 2019), Bayesian Optimization (Kandasamy et al., 2018; White et al., 2021a), and Evolutionary Algorithms (Lu et al., 2019; Sun et al., 2020b), among others Zoph et al. (2018); Wu et al. (2019).

However, a neural network's performance is contingent on both its architecture and weight configurations. Evaluating these architectures demands rigorous training and validation on specific datasets, which often leads to high computational costs, sometimes equivalent to thousands of GPU days (Zoph et al., 2018; Ying et al., 2019). In response to this challenge, recent research has focused on approach to streamline evaluations. For example, the weight-sharing approach (Cai et al., 2020; Chu et al., 2021) trains *super networks* which encapsulate all potential architectures, thereby abbreviating the training duration. A key advantage of super networks is the ability to inherit weights directly, obviating the need for redundant training. Another emerging approach involves the development of *predictors* (Liu et al., 2018; Wen et al., 2020; White et al., 2021a; Wei et al., 2022). Once calibrated on a curated subset of architectures, a predictor is expected to estimate the performance of architectures that have not been empirically tested.

In the realm of NAS predictors, the representations of architectures are pivotal. Recognizing neural architectures' inherent representation as Directed Acyclic Graphs (DAGs)(Ying et al., 2019), it is imperative to harness such topological information. Some research initiatives have gravitated towards

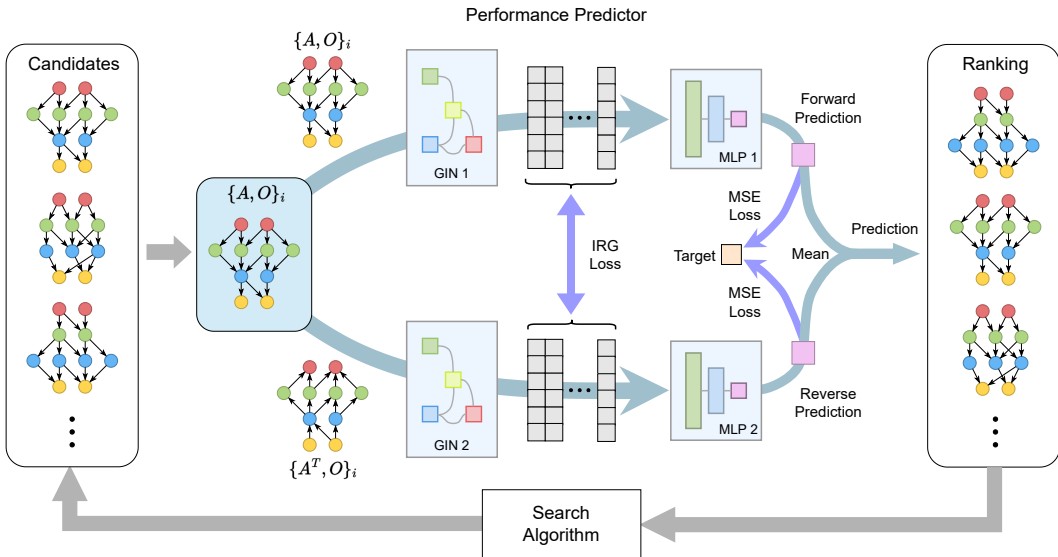

Figure 1: Illustration of our GNN predictor based NAS framework. Given architectures from the search algorithm, they are assessed by the performance predictor and represented using both forward and reverse graph encodings. These are processed by two GIN encoders into feature vectors, which are then fed to MLPs for prediction. To enhance model effectiveness, two training losses are incorporated, targeting both features and predictions.

serializing these DAGs(Sun et al., 2020a; White et al., 2021a). However, such serialized methods often provide an indirect or even incomplete representation, thus failing to encapsulate the gamut of topological intricacies. To bridge this gap, some research efforts has dedicated to *graph predictors*. The Graph Neural Networks (GNNs), including models such as the Graph Convolutional Networks (GCNs)(Kipf & Welling, 2016), Graph Isomorphism Networks (GINs)(Xu et al., 2019), and Graph Attention Networks (GATs) (Veličković et al., 2018), are specifically designed for processing graph-structured data. By iteratively aggregating neighborhood data, GNNs adeptly distill intricate feature representations for each vertex in a graph. This inherent competency has galvanized their integration within NAS frameworks, enabling GNNs to generate robust graph embeddings foundational to performance predictions. Empirical studies have shown their better predictive performance compared to sequence-based predictors (Chen et al., 2021b; Wei et al., 2022).

While GNN-based predictors have demonstrated effective performance, a limitation in most studies is the use of a *unidirectional* (forward) representation of the computation graph during predictor training (Liu et al., 2018; Jing et al., 2022; Wei et al., 2022). Neural architectures are inherently bidirectional, involving both forward and backward propagation phases. This raises the question: could leveraging this *bidirectionality* improve predictor performance? This research aims to address this question.

We initiate our study by repeatedly training a GNN predictor and analyzing the features extracted by encoders using visualization techniques. Our observations suggest that in the presence of limited training data, the encoder often faces challenges in effectively embedding features crucial for precise predictions. Given the shared valuable features across representations, it's logical to anticipate mutual refinement between the two encoders, enhancing data utilization. To tackle this challenge, we introduce a graph-based neural predictor that capitalizes on both forward and reverse graph representations. As illustrated in Fig. 1, every architecture is depicted as a DAG and its inverse. Both these representations feed into distinct GINs. To ensure that encoders from varied representations converge towards shared features, we incorporate a feature loss during the training phase at the embedding layer of the GINs. To encapsulate, our main contributions are:

- Through meticulous analysis of the GIN predictor outcomes employing both forward and reverse graph representations, we have highlighted relationships and distributions of fea-

tures across various search spaces. Our findings underscore the potential of dual graph representations in enhancing prediction accuracy.

- We have designed a performance predictor built upon a new network structure, where both forward and reverse graph depictions of architectures are employed. Moreover, we have introduced a tailored training loss to ensure congruence in embeddings generated by both GINs.

- Our comprehensive benchmark experiments juxtaposed our method against two state-of-the-art GNN-based NAS methods – NPNAS (Wen et al., 2020) and NPENAS (Wei et al., 2022). The experimental results, complemented by the ablation studies, ascertain the efficacy of our approach across multiple search spaces.

## 2 METHOD

In this section, we present the proposed FR-NAS method as illustrated in Fig. 1. We begin by outlining the architectural representations input to our predictor, followed by an in-depth exploration of our advanced encoder and predictor designs. Then, we introduce the tailored training loss, which is pivotal for the cohesive training of both the encoder and predictor, followed by some related empirical analysis.

### 2.1 ARCHITECTURE ENCODING

Since neural architectures can be generally conceptualized as DAGs, we adopt an adjacency matrix $\boldsymbol{A}$ to delineate edges connecting vertices and deploy a sequence of one-hot vectors $\boldsymbol{O}$ to represent operations at each vertex. The matrix $\boldsymbol{A} \in \{0,1\}^{N \times N}$ signifies a graph of $N$ vertices, where $\boldsymbol{A}_{i,j} = 1$ denotes an edge from vertex $i$ to vertex $j$. The one-hot encoding maps operations according to their sequence and generates a consistent-length vector for each vertex. Only one index with a value of 1 indicates the operation of the corresponding vertex. Specifically, we refer to his type of encoding to as *forward* graph encoding which propagates features in alignment with edge directions, while its *reverse* counterpart is derived by transposing the adjacency matrix to obtain $\boldsymbol{A}^T$.

### 2.2 ENCODER AND PREDICTOR

We integrate two distinct GINs to process the forward and reverse graph encodings, respectively. Specifically, each GIN encoder comprises three consecutive layers, where each layer employs dual fully connected structure with the ReLU activation function. Subsequently, a global mean pooling (GMP) layer extracts the embedding.

Within a singular GIN layer, every vertex accrues features from its preceding neighbors and its own data. This aggregated information then feeds into the ensuing fully connected layers. Due to the variant feature trajectory in the reversed graph, sharing weights becomes nonviable. Specifically, the encoding procedure of our proposed predictor can be articulated as:

$$\boldsymbol{h}_f = \text{Enc}(\boldsymbol{A}, \boldsymbol{O}; \boldsymbol{W}_1), \tag{1}$$

$$\boldsymbol{h}_r = \text{Enc}(\boldsymbol{A}, \boldsymbol{O}; \boldsymbol{W}_2), \tag{2}$$

where $\boldsymbol{A}$ is the adjacency matrix, $\boldsymbol{O}$ is the one-hot encoded operation sequence, $\boldsymbol{h}_f$ and $\boldsymbol{h}_r$ are the features embedded using forward and reverse directed graph data, and $\boldsymbol{W}_1$ and $\boldsymbol{W}_2$ are trainable weights in the encoder.

Following the encoding phase, we design a predictor derived from the embeddings. Given potential discrepancies between the initial embeddings from the encoder, deploying a unified predictor to process both embedded features may introduce biases. To mitigate this, we utilize two separate fully connected layers, each addressing the feature embeddings from a specific encoder. The ensuing prediction is ascertained by averaging the outcomes from the fully connected layers:

$$\boldsymbol{p}_f = \text{FC}(\boldsymbol{h}_f; \boldsymbol{W}_1), \tag{3}$$

$$\boldsymbol{p}_r = \text{FC}(\boldsymbol{h}_r; \boldsymbol{W}_2), \tag{4}$$

$$\boldsymbol{p} = (\boldsymbol{p}_f + \boldsymbol{p}_r)/2, \tag{5}$$

where $\boldsymbol{p}_f$ and $\boldsymbol{p}_r$ are the prediction results using forward and reverse graph data, FC is a sequence of two fully connected layers, and $\boldsymbol{p}$ is the final output of the predictor.

## 2.3 TRAINING LOSS

Our foremost training goal is to collaboratively refine both encoders with an emphasis on creating a harmonized representation strategy that promotes accurate predictions. With sparse training data, however, the encoders may not optimally capture pivotal features that significantly impact prediction accuracy (as discussed in Section 2.4). To mitigate this limitation, we propose a mutual learning strategy where the encoders reciprocally reinforce their ability to identify and exploit shared features.

In this regard, our training loss is meticulously formulated to bridge and diminish the discrepancies in the embedded features emanating from each encoder. However, a challenge emerges due to the inherent variability in the ordering of elements in the embedding vectors, making an element-wise comparison of encoder outputs unfeasible. Moreover, over-relying on the MSE loss for the output embedding vectors can be counterproductive, particularly when the encoders follow distinct information propagation mechanisms.

It is observed that neural architectures showcasing commensurate performance are likely to be underpinned by shared intrinsic features. Conversely, those manifesting divergent performance characteristics are discernibly disparate. Such intricate inter-relationships embedded within the architectural vectors not only encapsulate invaluable feature-related insights from each encoder, but are also robust against the capricious ordering of elements. Based on this observation and inspired by the instance relationship graph (IRG) (Liu et al., 2019b), we design a loss that minimizes the variance between the dual encoders as:

$$\mathcal{L}e = \frac{1}{M^2} \sum_{i=1}^{M} \sum_{j=1}^{M} (\|\boldsymbol{h}_{fi} - \boldsymbol{h}_{fj}\|_2^2 - \|\boldsymbol{h}_{ri} - \boldsymbol{h}_{rj}\|_2^2)^2, \tag{6}$$

where $\boldsymbol{h}_{fi}$ and $\boldsymbol{h}_{ri}$ symbolize features of the $i$-th architecture embedding emanating from the forward and reverse graph encoders, respectively. Notably, unlike the original IRG which learns existing knowledge from a teacher, our loss is designed to bidirectionally minimize the difference from scratch between the two encoders.

Subsequently, we employ a MSE of predictions and true performance as the prediction losses for the forward predictor ($\mathcal{L}_{pf}$) and reverse predictor ($\mathcal{L}_{pr}$) respectively:

$$\mathcal{L}_{pf} = \frac{1}{M} \sum_{i=1}^{M} (\boldsymbol{p}_{fi} - \boldsymbol{y}_i)^2, \tag{7}$$

$$\mathcal{L}_{pr} = \frac{1}{M} \sum_{i=1}^{M} (\boldsymbol{p}_{ri} - \boldsymbol{y}_i)^2, \tag{8}$$

where $M$ represents the number of architectures in one batch, $\boldsymbol{p}_{fi}$ and $\boldsymbol{p}_{ri}$ indicate the predicted performance of the $i$-th architecture, and $\boldsymbol{y}_i$ signifies the actual performance of the $i$-th architecture. Both $\mathcal{L}_{pf}$ and $\mathcal{L}_{pr}$ can be directly minimized as there is no interdependence between the two predictor components.

To culminate, the entire training loss is formulated as:

$$\mathcal{L}_1 = (1 - \lambda)\mathcal{L}_{pf} + \lambda\mathcal{L}_e, \tag{9}$$
$$\mathcal{L}_2 = (1 - \lambda)\mathcal{L}_{pr} + \lambda\mathcal{L}_e, \tag{10}$$

where $\lambda$ is a weight coefficient balancing the impact of the two losses. Given that the weights are intertwined due to multiple losses, we fine-tune the predictor iteratively. On each iteration, both $\mathcal{L}_1$ and $\mathcal{L}_2$ are determined individually to update the weights.

## 2.4 EMPIRICAL ANALYSIS

In this subsection, we focus on the training and evaluation of a straightforward graph-based predictor and delve into a detailed examination of the IRG matrix produced from the output embedding vectors of the encoders. Our analysis reveals the disadvantages of relying solely on one type of representation. This reliance often results in suboptimal embedding performance. In addition, we discuss the potential benefits of using two distinct representations to extract more detailed features.

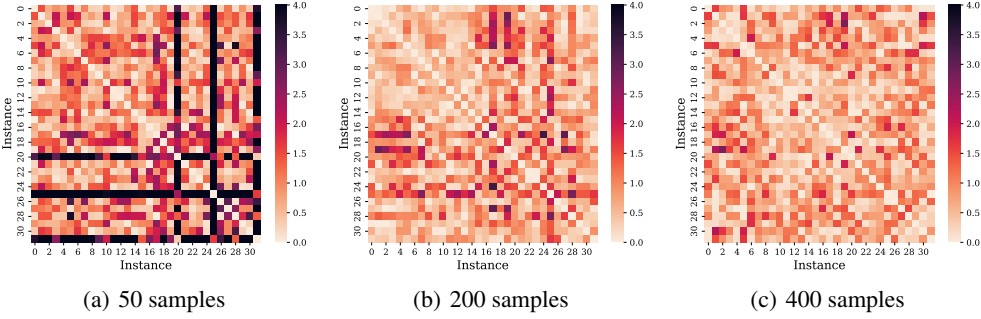

| (a) 50 samples | (b) 200 samples | (c) 400 samples |

Figure 2: Differences in the IRG matrix of embedding vectors when trained without the proposed feature loss, using 50, 200, and 400 samples from NAS-Bench-201.

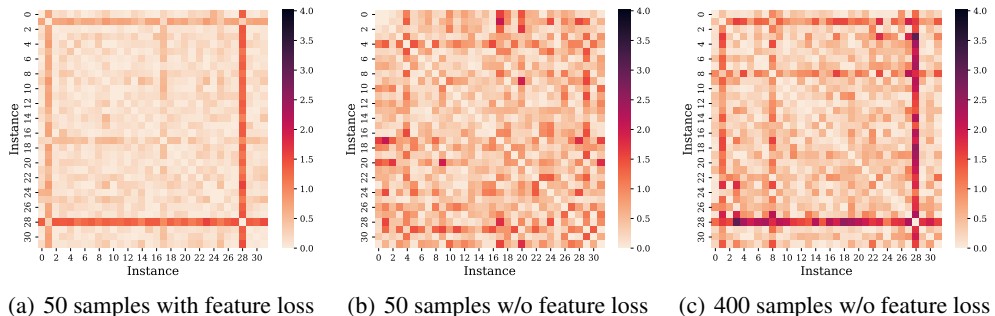

| (a) 50 samples with feature loss | (b) 50 samples w/o feature loss | (c) 400 samples w/o feature loss |

Figure 3: Differences in the IRG matrix of embedding vectors when trained with(out) the proposed feature loss, using 50 and 400 samples from DARTS search space.

We employed the GIN predictor on two distinct search spaces: the simpler NAS-Bench-201 (Dong & Yang, 2020) and the more complex DARTS (Liu et al., 2019a). After sampling 400 architectures at random, we divided them into groups of 50 and 200. This division allowed us to study the impact of increasing the volume of training data. Each predictor is trained from scratch on these subsets. Following this, we computed the IRG matrix for an additional set of 32 unseen architectures. From this matrix, we derived the element-wise distance matrix for both the encoders and decoders. The results are showcased in Fig. 2 and Fig. 3, where each data point on a heatmap represents the absolute difference in the distances encoded by the two separate encoders:

$$\text{Diff}(i,j) = \left| \|\boldsymbol{h}_{fi} - \boldsymbol{h}_{fj}\|_2^2 - \|\boldsymbol{h}_{ri} - \boldsymbol{h}_{rj}\|_2^2 \right|.$$

As indicated in Fig. 2(a), utilizing only 50 architectures for training results in significant differences between the two encoders. Given the constrained search space of NAS-Bench-201, expanding the training dataset reduces these discrepancies, as shown in Fig. 2(b). We also ran several trials with different subsets of the training data. We found a consistent pattern: in 42 out of 50 trials, there was a noticeable reduction in IRG loss as the sample size increased. This observation strengthens our belief that with adequate training data, both encoders tend to align in their knowledge. This alignment is particularly evident since they both perform identical regression tasks. However, in scenarios with limited data, the lack of comprehensive information affects both encoders. Combined with differing topologies, this can lead to changes in how information travels within the GNN layers. Consequently, the two GNN encoders might produce varied embeddings for the same architectures. Given that high-quality predictors often yield similar IRG matrices for subsequent predictions, a smart approach would be to synchronize the embeddings from both predictors. This synchronization should occur while simultaneously minimizing prediction errors to ensure the best features are captured.

The results in Figs. 3(a), (b), and (c) were obtained from the more challenging DARTS search space. For Fig. 3(a), we applied the feature loss method we introduced earlier but with a smaller data size. On the other hand, Figs. 3(b) and (c) show outcomes when trained with smaller larger datasets,

respectively, but without using the feature loss method. The 1st and 28th instances stand out as they represent unseen architectures with rare and high prediction errors. Predicting these unfamiliar architectures accurately is challenging, leading to disparities in the encoded distances compared to other instances. Without the feature loss method, and with limited training data, these subtleties are often missed. In contrast, our proposed technique effectively pinpoints these detailed features, even when the training dataset is smaller.

## 3 EXPERIMENTS

In this section, we measure the performance on three benchmark search spaces: NAS-Bench-101 (Ying et al., 2019), NAS-Bench-201 (Dong & Yang, 2020), and the DARTS search space. First, we introduce the settings of the search space, baseline algorithms, and hyperparameters. Then, we present the overall performance results on different training data sizes and search spaces. Finally, we perform a comprehensive ablation study to demonstrate the effectiveness of our methods.

### 3.1 EXPERIMENT SETUP

#### 3.1.1 SEARCH SPACES

Operations are represented on nodes, which can be directly represented using our architecture representation. The NAS-Bench-101 search space is a cell-based space consisting of 423k unique convolutional architectures. These architectures are trained and tested on CIFAR-10, providing ample labeled data for predictor training. Three types of operations are applied to operation vertices. Unlike NAS-Bench-101, the original representation for NAS-Bench-201 and Darts represents operations on edges and gathers features on nodes. We modify the representation such that operations are on vertices. Additionally, since the DARTS search space comprises approximately $10^{21}$ architectures, we directly sample from the training data rather than the proxy data of predictors from NAS-Bench-301(Zela et al., 2022). In this experiment, we use validation accuracy on CIFAR-10 as our training data. For all labeled data, we convert the validation accuracy into an error percentage. Experiments are conducted using benchmark data from the EvoXBench (Lu et al., 2022) database.

#### 3.1.2 PEER METHODS

We evaluate our proposed FR-NAS in comparison with two state-of-the-art peer methods: NPE-NAS (Wei et al., 2022) and NPNAS (Wen et al., 2020), both of which are NAS methods using GNN-based predictors. We usethe Kendall rank correlation coefficient $\tau$ as the metric. As NAS aims to identify architectures with superior performance, ranking accuracy is crucial for performance prediction. Higher ranking accuracy increases the likelihood of the NAS algorithm finding optimal architectures. NPNAS employs GCN as a predictor and introduces a directed GCN layer to address the limitations of GCN on directed graphs. This layer comprises two GCN layers: one taking forward input and the other reverse input. The output features of both inputs are then averaged to produce the final output. Since it integrates both forward and reverse graphs in a single layer, we cannot apply our framework to such a GCN encoder. We compare the results of the NPNAS predictor with our proposed FR-NAS predictor to demonstrate the advantages of utilizing two representations. The GIN predictor from NPENAS highlights the capabilities of GIN layers in architecture representation on directed graphs and has shown superior performance. For fair comparisons, we employ the same data preprocessing method as in the original NPENAS.

#### 3.1.3 HYPERPARAMETERS

The only additional parameter in our approach is the weight coefficient $\lambda$, which is set to 0.8 according to our parameter sensitivity analysis in Fig. 4. For fair comparisons, we maintained the same hyperparameters and the structure of the two GIN predictors as in the single GIN predictor of NPENAS. We also implemented the GCN predictor using settings consistent with NPNAS. The hidden size of the two fully connected layers within the GIN layer is set to 32, except for the first GIN layer. This first layer is designed to handle the one-hot feature, which has dimensions of 6 for NAS-Bench-101, 7 for NAS-Bench-201 and 13 for DARTS, while the prediction layer employs a hidden size of 16. When training the GIN predictor for both NPENAS and FR-NAS, batch normal-

Table 1: Comparison of Kendall $\tau$ correlation for predictors trained with varying data sizes across NAS-Bench-101, NAS-Bench-201, and DARTS search spaces. Values represent the mean Kendall $\tau$ over 200 runs, with the best result in each section highlighted in bold.

| Search Space | Algorithm | Training Data Size | | | | | |
|---|---|---|---|---|---|---|---|
| | | 50 | 100 | 150 | 200 | 300 | 400 |
| NAS-Bench -101 | NPNAS | 0.5499 | 0.5636 | 0.5812 | 0.5924 | 0.6138 | 0.6309 |
| | NPENAS | 0.4470 | 0.5692 | 0.6196 | 0.6444 | 0.6755 | 0.6942 |
| | FR-NAS (ours) | **0.5556** | **0.6596** | **0.6950** | **0.7131** | **0.7359** | **0.7496** |
| NAS-Bench -201 | NPNAS | 0.4810 | 0.5221 | 0.5538 | 0.5865 | 0.6202 | 0.6400 |
| | NPENAS | 0.5587 | 0.6560 | 0.6983 | 0.7266 | 0.7631 | 0.7882 |
| | FR-NAS (ours) | **0.6305** | **0.7129** | **0.7476** | **0.7730** | **0.8061** | **0.8262** |
| DARTS | NPNAS | 0.4813 | 0.5568 | 0.5934 | 0.6068 | 0.6254 | 0.6389 |
| | NPENAS | 0.4656 | 0.5409 | 0.5691 | 0.5851 | 0.6125 | 0.6275 |
| | FR-NAS (ours) | **0.5334** | **0.5989** | **0.6237** | **0.6425** | **0.6666** | **0.6818** |

ization is added after each GIN block and fully connected layer. A dropout layer, with a dropout rate of 0.1, is placed after the GIN encoder's embedding.

For optimization, we utilize the Adam optimizer with a cosine annealing learning rate, where the initial learning rate and weight decay are set at $5 \times 10^{-3}$ and $1 \times 10^{-4}$, respectively. We use data sets ranging in size from 50 to 400 for training, while testing is conducted with a data set of 5,000 samples. Training of the GIN predictor lasts for 300 epochs with a batch size of 16. To reduce potential biases from sampling, 200 independent random experiments were conducted.

## 3.2 RESULTS

The experimental results are summarized in Table1. Generally, our method outperforms both peer methods in all cases. The GIN predictor from NPENAS excels with larger data sizes. While the GIN predictor employs an MLP to aggregate representations, it can assimilate more information. By contrast, the GCN predictor outperforms the GIN predictor with a training data size of 50 in the larger search spaces, indicating a substantial performance decline for the GIN predictors. Using the same GIN encoder settings, our method enhances performance, especially with smaller data sizes.

Figure 4: Parameter sensitivity analysis of the weight coefficient $\lambda$.

## 3.3 ABLATION STUDY

There are two pivotal mechanisms in the FR-NAS predictor for performance improvements: (1) the two GIN encoders taking forward and reverse graph as input; (2) the feature loss as well as the original prediction loss. To evaluate the effectiveness of the two mechanisms, we compare FR-NAS with some variants of NPE-NAS for ablations study: a Single-direction ensemble using the same architecture as FR-NAS but taking only the forward direction of graph encoding as inputs (denoted as NPENAS-Forward), a forward-and-reverse ensemble using the same input and architecture as our method (denoted as NPENAS-FR).

As shown by the results in Fig. 5, a simple forward and reverse ensemble predictor can be better than ensembles of single direction predictor. Moreover, when applying the new training loss from our method, the performance is generally improved, especially when the training data size is small. This indicates that such structure can better utilize two representations.

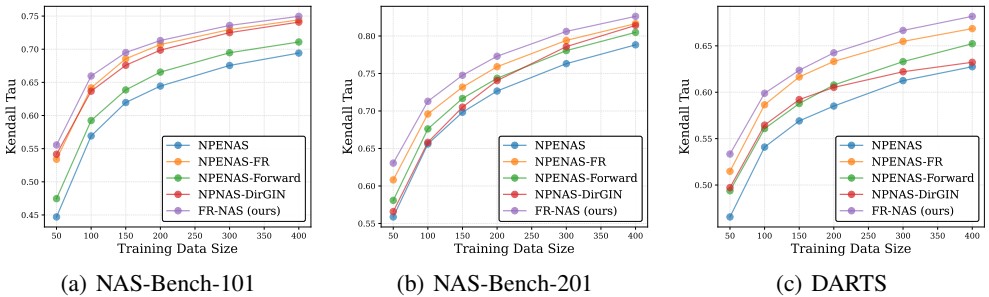

(a) NAS-Bench-101     (b) NAS-Bench-201     (c) DARTS

Figure 5: Comparison of variants on three benchmark datasets. NPENAS-Forward: employing the same architecture as FR-NAS but taking only the forward direction of graph encoding as inputs. NPENAS-FR: utilizing the same inputs and architecture as FR-NAS but without using our training approach. NPNAS-DirGIN: a GIN adaptation of NPNAS featuring bidirectional GIN layers.

To further demonstrate the efficacy of our method, we construct a directed GIN predictor based on the GCN variant as introduced by NPNAS, which we term as NPNAS-DirGIN. It is noteworthy that NPNAS inherently considers both the forward and reverse graph configurations on a layer-wise basis. A comparative analysis presented in Table 1 indicates that our method yields significantly improved outcomes. However, such comparison arises an ambiguity. Given that distinct GNN layers are applied for our method and NPNAS, it remains uncertain whether both representations can be efficiently harnessed by the mechanism of NPNAS. To address this, we tailored a directed GIN layer that amalgamates the outputs of two GIN layers: one emphasizing the forward propagation of information and the other focusing on the reverse. Preliminary results for NPNAS-DirGIN in Figs. 5 (a)(b) suggest that this combined layer aligns most closely with the outcomes from individual forward and reverse ensembles within smaller search realms, while it is evident that the mechanisms adopted by NPNAS might not align seamlessly with GIN layers. This incongruity potentially arises since there is no dedicated mechanism ensuring that both facets of a directed GIN layer assimilate meaningful features during training. In expansive search scenarios like those with DARTS in Fig. 5(c), there is a diminished likelihood of gleaning additional insights from dual representations.

# 4 RELATED WORK

This section provides a brief review of the existing literature on GNN-based performance predictors, techniques for utilizing multiple data sources, and contrasts with the domain of contrastive learning.

## 4.1 PERFORMANCE PREDICTORS

At a fundamental level, neural networks are visualized as computational graphs. Various strategies have been proposed to encode these architectures, with some approaches opting for serialized encodings. However, these often struggle with maintaining topological intricacies (Liu et al., 2018). BANANAS (White et al., 2021a) suggests a meticulous path-based encoding scheme. Due to GNNs' inherent ability to handle graph data, they become ideal for capturing both topological structures and associated operations. Current research largely employs GNNs for extracting embeddings from graph-encoded data for subsequent performance prediction. For instance, NPNAS (Wen et al., 2020) utilizes the graph convolutional network (GCN) (Kipf & Welling, 2016), while HOP (Chen et al., 2021b) advocates for the graph attention network (GAT) (Veličković et al., 2018). NPENAS (Wei et al., 2022) employs the graph isomorphism network (GIN) (Xu et al., 2019) and isolated node representation, which has demonstrated superiority in handling directed graphs. Despite the occasional inclusion of reverse graph data, the general trend seems to favor the development of more powerful predictors primarily oriented around forward graph data.

## 4.2 DATA AUGMENTATION

The field of data augmentation continues to innovate, introducing techniques to bolster predictor performance. HAAP Liu et al. (2021), for example, employs homogenized representations transformed into sequences. This stands in contrast to our method, which maps distinct graph representations to identical labels. Methods like OMNI (White et al., 2021b) combine various data sources, while others like FBNetV3 (Dai et al., 2021) and GMAE (Jing et al., 2022) utilize auxiliary information or modified graph data. An area yet to be thoroughly explored is the potential of the reverse graph as an alternative data source, a direction which holds promise for producing rich augmentations.

## 4.3 CONTRASTIVE LEARNING

Contrastive learning has emerged as a prominent technique for producing robust feature embeddings, especially when data is scarce. For instance, GraphCL (You et al., 2020) presents a graph contrastive framework where perturbation techniques generate variant graphs which are then processed by a GNN encoder. In the context of NAS, CTNAS (Chen et al., 2021a) assesses the relative performance of diverse architectures using GCNs. Another recent approach Hesslow & Poli (2021) leverages an extended data jacobian matrix for a contrastive network. Distinctively, our method employs different graph encodings for the same architectural input. Unlike existing methods that focus on varying inputs to a single network, ours centers on dual GNN encoders handling different encodings.

## 5 CONCLUSION

In this research, we introduced an advanced GNN predictor that simultaneously leverages forward and reverse graph representations. By delving into the intricacies of GNN predictors and understanding the impact of various graph representations on encoding and prediction, we identified the merit of bidirectional topological consideration. This insight suggested potential enhancements in prediction accuracy. Our unique training loss, which is tailored to optimize feature extraction, further supported this hypothesis. Experimental results not only validated our method but also highlighted a marked performance edge over traditional GNN predictors. Notably, our research offers strategies to amplify the effectiveness of GNN predictors, particularly in data-limited settings.

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
