# OpenReview forum: "FR-NAS: Forward-and-Reverse Graph Predictor for Efficient Neural Architecture Search"
_ICLR.cc/2024/Conference — ICLR 2024 Conference Withdrawn Submission_

### Official Review · Reviewer_YbEc · 2023-10-30

**Soundness:** 2 fair
**Presentation:** 2 fair
**Contribution:** 2 fair
**Rating:** 5
**Confidence:** 4

**Summary:**

This paper proposed a GNN-based performance predictor for NAS, the bidirectionality information is employed for performance improvement, and some experiments are conducted for verification.

**Strengths:**

Using the bidirectionality information to improve the performance of the predictor is quite interesting because almost all existing works did not recognize this point.

**Weaknesses:**

The peer competitors used for comparison in this paper are not SOTA. This paper should not only compare the methods based on GNN but also SOTA performance predictors based on other techniques.

The experiments should also go to ImageNet, instead of only the measures for performance predictors, the final goal of which is for NAS.

**Questions:**

See Above

---

### Official Review · Reviewer_NESX · 2023-10-30

**Soundness:** 3 good
**Presentation:** 3 good
**Contribution:** 3 good
**Rating:** 5
**Confidence:** 4

**Summary:**

The FR-NAS paper devised a new graph neural network (GNN) based surrogate model for neural architecture search. The adjacency matrix is passed to a GNN which encodes the forward propagation of the neural network. Its transpose is passed to another GNN which encodes the backward propagation. These encodings are passed to their respective predictors ($p_{f}$ and $p_{r}$) and the final predictor is an average of these two. $p_{f}$ and $p_{r}$ are trained using mean squared error loss between the predicted and the true accuracies of the networks. To ensure that the forward and backward embeddings are consistent with each other, they used an additional loss to enforce that the relative distance between two architectures in the forward embedding space and the backward embedding space is similar.

 They report the results on NasBench-101, NasBench-201 and Darts search space. In their ablation studies, they bolster the case for using both the forward and the backward pass encodings.

**Strengths:**

1. Using two encoders to capture the forward and backward propagation encodings and using the IRG loss to synchronize them is novel.
2. Their algorithm outperforms the other 2 baselines on NASBench-101, NASBench-201 and the DARTS search space.

**Weaknesses:**

1. Please compare against [1], [2] which are also GCN based predictors.
2. It is also important to demonstrate that the surrogate model is competitive to other baselines such as those included in Neural architecture optimization (NAO) [3], BANANAS [4] and other predictors in  [5]
3. For all the baselines, please report the time taken to train the predictors and to compute the correlations on all the 3 benchmarks.

[1] BRP-NAS: Prediction-based NAS using GCNs,  Dudziak et al.
[2] Bridging the gap between sample-based and one-shot neural architecture search with bonas, Shi et al.
[3] Neural Architecture Optimization, Luo et al.
[4] BANANAS: Bayesian Optimization with Neural Architectures for Neural Architecture Search, White et al.
[5] How Powerful are Performance Predictors in Neural Architecture Search? White et al.

**Questions:**

1. Can you please tabulate figure 5? Given that the NPENAS-FR and FR-NAS plots are very close to each other as the training data increases, it would be good to see the actual correlation values.
2.Did you consider other alternatives to the feature loss?  Given that both $L_{pf}$ and $L_{pr}$ are predicting the accuracy of the same architecture, what would happen if you minimize the divergence between the outputs of $L_{pf}$ and $L_{pr}$ predictors?
3. Given that the algorithm is trained to minimize the feature loss, it would have the least Diff(i,j) when compared to those that are trained without them. So is figure 3 a fair comparison?

---

### Official Review · Reviewer_6p2a · 2023-10-31

**Soundness:** 2 fair
**Presentation:** 2 fair
**Contribution:** 1 poor
**Rating:** 3
**Confidence:** 5

**Summary:**

This paper proposes FR-NAS, a neural architecture performance predictors that estimates performance using both the forward-pass and backwards-pass representation of a NAS architecture. FR-NAS uses an Instance Relation Graph (IRG) loss to train the dual encoder. The author's evaluate the method on three NAS-Benchmarks and compare to known predictors NPENAS and NPNAS, outperforming both.

**Strengths:**

There is some novelty to considering the backwards pass representation of a NAS architecture when making a prediction.
The evaluation shows that FR-NAS conclusively defeats NPNAS and NPENAS on NAS-Bench-{101, 201, 301} at every training dataset size.
There are additional ablation studies for some components.

**Weaknesses:**

The novelty of this work is somewhat limited as its really only using a dual encoder with adjacency matrix transpose, while components like the IGR loss are heavily borrowed from different work.

This work only considers experiments on cell-based NAS Benchmarks but not on real NAS problems, which are outperformed by macro-based NAS structures like Once-for-All/MobileNets/EfficientNets. Also, no search is applied and no found architectures are evaluated.

There are probably simpler ways to consider the backwards-pass representation of a NAS DAG which this paper does not consider. The trivial method would be to simply cast the DAG as a fully-directed graph (for every edge (i, j), add edge (j, i)) and still use a simpler encoder. Another way, also simpler than this would be to consider weighted edges, e.g., forward-pass edges have weight '1', backwards-pass have weight '-1', and you use torch.nn.GINEConv instead of torch.nn.GINConv.

The IGR loss in this paper is somewhat counter-intuitive. The intuition behind considering the transposed adjacency matrix is that you are providing the predictor with new information not found in the forward-pass adj. matrix. This information should allow the predictor to learn a better understanding of architecture performance, which would help performance. Under this assumption, you would expect the encodings of the forward encode and backwards encoder to probably be different as they should be learning on distinct information, and that the concatenation of that information (graph embeddings from each encoder) benefits predictor performance. Instead, the IGR loss is counter to this as it forces both the forward/backwards encoders to 'learn the same thing' using different views of the same data. In other words, in Figs 2-3, showing how the IRG matrix values goes down with the addition of the loss and more samples seems counter-intuitive.

Analysis in Figures 2 and 3 is missing NAS-Bench-201 using the IGR loss, NAS-Bench-301 400 samples with the loss, even though the manuscript is not even 9 pages.

Author's mention Graph Attention Networks (GAT) a few times in the paper, but do not use them to perform any analysis on their encoders, e.g., highlighting the nodes/edges assigned high attention scores. This would be a good way to highlight how their method learns and the benefits of their design, and rebut the hypothesis that FR-NAS outperforms NPNAS and NPENAS simply because the predictor has more parameters.

There are a lot of missing entries in the related work section. Some of which should be added, and compared to, e.g.,
- TNASP [1] and PINAT [2] deal with special encodings for the adjacency matrix, like this paper.
- CDP [3] is a cross-domain predictor which cases 201 and 301 to be like 101, to deal with limited target data like this paper.
- GENNAPE [4] also deal with limited target data like CDP, but they also utilize a robust form of Computational Graph that covers the entire architecture, not just the NAS cell design.
- Multi-Predict [5] show how to leverage other information like Zero-Cost Proxies [6] and device latency/FLOPs to aid prediction - both of which this paper does not acknowledge, yet it is a critical concern of NAS.

For the above reasons I would recommend rejection of this manuscript.

References:

[1] Lu et al., "TNASP: A Transformer-based NAS Predictor with a Self-Evolution Framework", in NeurIPS 2021.

[2] Lu et al., "PINAT: A Permutation INvariance Augmented Transformer for NAS Predictor", in AAAI-23.

[3] Liu et al., "Bridge the Gap Between Architecture Spaces via a Cross-Domain Predictor", in NeurIPS 2022.

[4] Mills et al., "GENNAPE: Towards Generalized Neural Architecture Performance Estimators", in AAAI-23.

[5] Akhauri and Abdelfattah, "Multi-Predict: Few Shot Predictors For Efficient Neural Architecture Search", in AutoML Conf 2023.

[6] Abdelfattah et al., "Zero-Cost Proxies for Lightweight NAS", in ICLR 2021.

**Questions:**

Not a question but minor nitpick: Eq. 2 should be Enc(A_{T}, O; W_2)

---

### Official Review · Reviewer_wK9q · 2023-11-10

**Soundness:** 3 good
**Presentation:** 3 good
**Contribution:** 2 fair
**Rating:** 3
**Confidence:** 5

**Summary:**

This paper proposes a new GNN performance predictor for NAS that considers the forward and reverse computational graph of architectures. Furthermore, the authors also propose a loss function that minimizes the variance between the dual encodings of the forward and backward pass. Experiments in standard tabular and surrogate benchmarks show improvements NPNAS and NPENAS.

**Strengths:**

- The paper presents a simple and effective way to improve the predictive performance of GNNs for NAS. The empirical evaluation demonstrates that the performance increases with the number of datapoints, which is nice to see.

- Easy to read and clearly written.

- Compared to NPNAS and NPENAS, the proposed algorithm shows significant improvements.

**Weaknesses:**

- The proposed method to encode both the forward and backwards encoding is well-known in literature (see section 3.4 in [1] for instance) as well as in NAS [2]. The linear scalarization of the prediction loss with the loss term that minimizes the variance between the two encoders is trivial.

- The authors evaluate their method on 3 tabular/surrogate benchmarks. I think this is not enough considering the diversity of available NAS benchmarks out there. There are more interesting NAS benchmarks (see NAS-Bench-Suite [3]) that also have evaluated NPENAS, and therefore makes the comparison to the proposed method possible.

- No code available at submission time.


**References**

[1] https://arxiv.org/pdf/1904.11088.pdf

[2] https://arxiv.org/pdf/2010.04683.pdf

[3] https://arxiv.org/pdf/2201.13396.pdf

**Questions:**

- Can the authors evaluate their method on the same framework as used in [4]? It would be great to see how FR-NAS performs under the same settings as those methods are evaluated.

- What is the performance of the predictor inside a NAS algorithm? Can you evaluate FR-NAS as done in NAS-Bench-Suite (see Table 2)?

**References**

[4] https://arxiv.org/pdf/2104.01177.pdf